# Applications and Limitations of Quantifying Speciated and Source-Apportioned VOCs with Metal Oxide Sensors

**Kristen Okorn [1,]\* and Michael Hannigan [2]**

[1] Environmental Engineering, University of Colorado Boulder, Boulder, CO 80309, USA
[2] Mechanical Engineering, University of Colorado Boulder, Boulder, CO 80309, USA; michael.hannigan@colorado.edu
[\*] Correspondence: kristen.okorn@colorado.edu

**Abstract:** While low-cost air quality sensor quantification has improved tremendously in recent years, speciated hydrocarbons have received little attention beyond total lumped volatile organic compounds (VOCs) or total non-methane hydrocarbons (TNMHCs). In this work, we attempt to use two broad response metal oxide VOC sensors to quantify a host of speciated hydrocarbons as well as smaller groups of hydrocarbons thought to be emanating from the same source or sources. For sensors deployed near oil and gas facilities, we utilize artificial neural networks (ANNs) to calibrate our low-cost sensor signals to regulatory-grade measurements of benzene, toluene, and formaldehyde. We also use positive matrix factorization (PMF) to group these hydrocarbons along with others by source, such as wet and dry components of oil and gas operations. The two locations studied here had different sets of reference hydrocarbon species measurements available, helping us determine which specific hydrocarbons and VOC mixtures are best suited for this approach. Calibration fits on the upper end reach above $R^2$ values of 0.6 despite the parts per billion (ppb) concentration ranges of each, which are magnitudes below the manufacturer's prescribed detection limits for the sensors. The sensors generally captured the baseline trends in the data, but failed to quantitatively estimate larger spikes that occurred intermittently. While compounds with high variability were not suited for this method, its success with several of the compounds studied represents a crucial first step in low-cost VOC speciation. This work has important implications in improving our understanding of the links between health and environment, as different hydrocarbons will have varied consequences in the human body and atmosphere.

**Keywords:** low-cost sensors; volatile organic compounds; positive matrix factorization; source attribution; VOC speciation

## 1. Introduction

### 1.1. The Need for Low-Cost Speciated Hydrocarbon Data

While numerous studies have utilized low-cost sensors to study generalized VOCs or TNMHCs in addition to methane [1–4], few have attempted to characterize specific hydrocarbons other than methane in outdoor monitoring experiments. These current characterizations have limited utility; it is helpful to know when and where hydrocarbon emissions are significant, but not knowing the exact chemical composition leaves lingering questions about source attribution as well as impacts on climate and human health, since some hydrocarbons are more damaging to people and the environment than others [5,6]. Additionally, since VOC sources can be exceptionally small or transient, outdoor VOC hotspot detection is an up-and-coming area in exposure assessment [7,8]. Likewise, VOC source attribution is critical not only for individuals to understand their exposure risk, but also in community planning and mitigation efforts. Since myriad VOC sources, including oil and gas operations, may exist within a community, it is crucial to disentangle them to determine which emit the highest concentrations or most harmful compounds so that

risks can be addressed by local organizations in order of importance [9–12]. This also has important implications on secondary pollutants generated in the troposphere, as the exact hydrocarbon makeup of emissions will determine their subsequent reaction pathways and byproducts [13].

While low-cost sensors are available for speciated hydrocarbons such as formaldehyde (HCHO), several weaknesses have prevented them from becoming more widely used. As many hydrocarbon-specific sensors have been developed for indoor use [14–17], years of research and trial and error will be required to determine which—if any—are suitable for long-term ambient monitoring in outdoor environments. Using a formaldehyde-specific sensor also closes the door to studying other types of hydrocarbons. If there are 30 hydrocarbons of interest, sensor packages do not currently have the capability to run 30 species-selective sensors all at once; selecting a few means neglecting the rest. For these reasons, we have opted to use two generalized VOC sensors to quantify a host of speciated hydrocarbons as well as groups of hydrocarbons. This method is imperfect; most commercially available sensors struggle with detection limits higher than the observed concentrations of individual VOCs and lack selectivity towards certain hydrocarbons [18]. However, this approach still allows us to view hydrocarbons in totality rather than only quantifying a select few.

At the regulatory level, summa cannisters and gas chromatography (GC) are typically used to determine the concentrations of speciated VOCs to a high degree of accuracy. Field-deployable continuous mass spectrometry data used in this work have associated uncertainties in the parts per billion range if not the mere parts per trillion (ppt). While these techniques deliver high-quality data, they are expensive to own and operate, with monthly operating costs typically totaling in the thousands [19,20]. Thus, utilizing this data only for sensor calibration purposes rather than entire deployments would greatly reduce operating expenses. Likewise, at the current instrument price point, multiple measurement locations cannot be established. Using low-cost sensors instead, we gain the ability to measure speciated hydrocarbons at multiple locations throughout an area of interest, highlighting localized differences on the relevant spatial and temporal scales.

### 1.2. VOC Quantification Using Low-Cost Sensors

One previous study used the same two generalized VOC sensors—the Figaro TGS 2600 and 2602—to quantify speciated VOC data [21]. As multivariate linear regression has been frequently used to match low-cost sensor signals to reference concentrations [1–3,18,21–25], Collier-Oxandale and co-workers used this method to quantify benzene as an individual hydrocarbon as well as aromatics as a group (benzene, toluene, $C_8$, and $C_9$). Using one of the pod datasets explored in this work, $R^2$ values between the reference and sensor benzene signals reached 0.72 for calibration and 0.67 for validation, explaining nearly three-fourths of the variability in the species. In Collier-Oxandale's iteration, carbon dioxide data from a reference instrument were incorporated into the multivariate linear regression along with the low-cost sensor signals to improve fits [21]. However, since additional reference signals are infrequently available, we will focus our analysis on achieving similar fits without the aid of any additional reference signals, relying solely on the sensors. We believe this is a more realistic approach as we step away from frequent, expensive calibration campaigns. Instead, we explore a less commonly used machine learning technique to improve fits. We also consider a different hydrocarbon grouping technique, this time by source as described in Section 2.3 rather than by physical structure as employed previously by Collier-Oxandale and co-workers.

Other previous low-cost sensor works used both multivariate linear regression and artificial neural networks (ANNs) to quantify several more commonly studied trace gases, including ozone, methane, carbon monoxide, carbon dioxide, and nitrogen dioxide in the ambient environment [4,26]. ANNs routinely outperformed linear models, and the varying input sensor measurements seemed to have a greater impact on the goodness of fit than the parameter tweaking performed. Early stopping also proved to be a valuable tool in

preventing overfitting [26]. ANNs have also been successful in quantifying speciated VOCs from metal oxide sensors in a laboratory environment [27–29]. This work will be among the first at the intersection of these two categories: using ANNs to quantify speciated VOCs in an outdoor environment, opting for a field normalization technique rather than a chamber calibration.

### 1.3. Application to Oil and Gas Data

The sensor locations utilized in this work are all near oil and gas facilities on Colorado's Front Range, making this one of the main sources of hydrocarbons in each area. As each facility pulls from the Wattenburg Oilfield [30], the chemical composition of associated hydrocarbon gas mixture may include similarities at each location. However, as the extraction and refining activities at each site vary, site to site differences in emissions from the same oilfield are likely. As further discussed in the instrumentation section, the hydrocarbons studied at each site varied as the reference measurement methods varied at each. Thus, our comparisons are further muddled as only two of the same pollutants were available at each site, with the rest remaining unique to their deployments.

## 2. Methods

Here, we make use of low-cost sensor data collected at regulatory monitoring sites near oil and gas facilities during two distinct campaigns. In the summer of 2014, we set up our monitors at a NASA-operated facility measuring VOCs as part of their Front Range Air Pollution and Photochemistry Experiment (FRAPPE) for approximately one month. Years later in 2020, we co-located our monitors at another VOC measurement site managed by the Colorado Department of Public Health and Environment (CDPHE) for approximately a month towards the end of the summer. In this work, we also apply our calibration fits from the CDPHE site to a week of field data collected on the grounds of an oil and gas facility in Eastern Colorado. Figure 1 shows each of the pod locations and durations with respect to oil and gas wells in the area as provided by the Colorado Oil and Gas Conservation Commission (COGCC) [30].

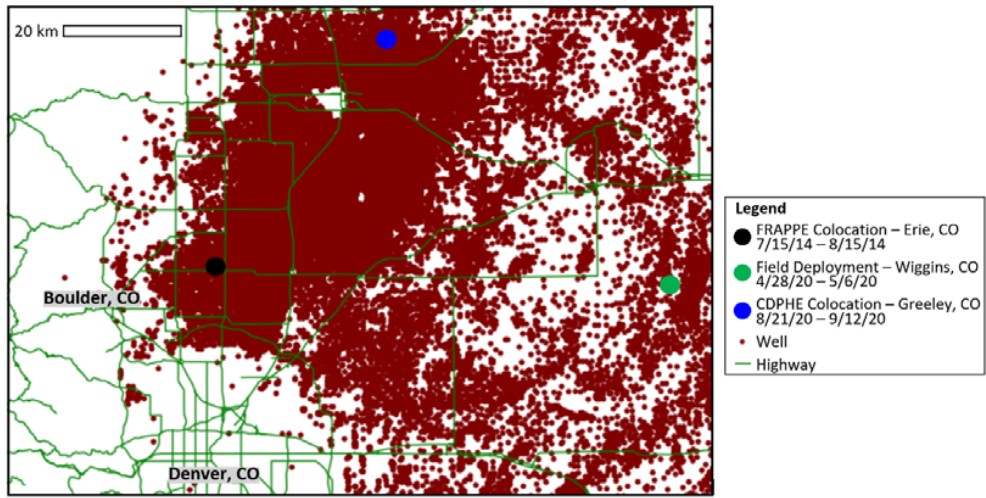

**Figure 1.** Map of colocation and deployment locations relative to oil wells.

### 2.1. Instrumentation

Low-cost air quality monitors dubbed U-Pods and Y-Pods [3,23], two generations of the same pod platform, were used for data collection. Both are built in-house by the Hannigan lab and include a suite of environmental and gas-phase sensors, including temperature, pressure, $CO_2$, and two metal oxide hydrocarbon sensors (TGS 2600 and 2602, Figaro USA Inc., Arlington Heights, IL, USA). At the FRAPPE site in Erie, CO, the pod was mounted to the railing of a 1-story trailer, approximately 1 m away from the reference instrument inlet [21]. At the CDPHE site in Greeley, CO, USA, pods were stacked on top of

a container on the roof of the trailer within a meter of the reference instrument inlet. Note that during the colocation, the container was knocked over by the wind, so by the end of the colocation, the pods were not as close to the inlet as desired. For this reason, we have elected to show the data from only one pod, which we believe was the best positioned despite the stack toppling over.

Reference-grade mass spectrometry instruments were also utilized at each calibration site. For the FRAPPE colocation in Erie, CO, USA, a proton transfer reaction mass spectrometry (PTRMS) was operated as part of NASA FRAPPE [31]. At the site operated by CDPHE in Greeley, CO, USA a gas chromatography mass spectrometry (GCMS) instrument was utilized for speciated hydrocarbon measurements. Note that while minute-average data were available from the FRAPPE campaign, the data resolution for the CDPHE data was approximately one measurement per 45 min; we re-timed this to hourly based on the median timestamp and used hourly averages for our sensors as well.

### 2.2. Artificial Neural Networks

Based on previous success in using ANNs to quantify pollutants with low-cost sensors [26], we focused our efforts on this machine learning technique, which has not been used previously for this specific hydrocarbon application. When using multivariate linear regression to quantify pollutants such as methane and generalized VOCs previously, interaction terms among sensors proved beneficial [1,2,21]. For instance, the ratio or product of the two VOC sensors has been added as an additional term to better quantify methane. Trial and error was used to determine the best combination of input factors, starting with electrical signal: raw voltage in mV, converted resistance in $m\Omega$, or normalized resistance ($R/R_0$) as recommended by the manufacturer. Then, additional sensor signals such as $CO_2$ and interaction terms among the two metal oxide sensors, temperature, and humidity were added to determine the best combination. Finally, ANN parameters such as the learning rate and number of epochs were tuned using trial and error based on literature recommendations [4,26,28]. Data were split into calibration and validation randomly using a three-fold algorithm, and early stopping was used by default to help to prevent overfitting.

### 2.3. Positive Matrix Factorization

In addition to quantifying individual hydrocarbons, we sought to group hydrocarbons by their source. This aids in our understanding of which pollutants are emitted by the same source and may also improve fits; summed hydrocarbons will have higher concentrations than individual species, raising the total concentration closer to the sensors' prescribed detection limits. These source attribution efforts afford the relevant communities or agencies the ability to address hazards in order of importance. For instance, a previous PMF analysis near an oil and gas facility in Los Angeles resulted in VOCs split into 6 main factors, with compounds emanating from: combustion, aged motor vehicle emissions, fresh motor vehicle emissions, biogenic, industrial manufacturing, and natural gas [32]. This analysis aided in community understanding of anthropogenic and biogenic air quality risks in the area.

Although certain compounds may be emitted from more than one source (e.g., styrene can be found in both industrial and oil and gas emissions), source attribution in this manner affords us a more compartmentalized view of often overwhelming and tangled webs of VOC species present in a single area. In this aim, we utilized the EPA PMF 5.0 tool, a mathematical receptor model, which essentially categorizes input concentrations by their likely sources and contributions [33]. PMF is governed by Equation (1).

Equation (1): PMF Governing Equation, where $X_{ij}$ is the input data matrix of ambient concentrations (*i* samples and *j* compounds), *p* is the number of factors as designated by the user, *g* is each factor's contribution, *f* is the chemical profile, and $e_{ij}$ is the residual error.

$$X_{ij} = \sum_{k=1}^{p} g_{ik}f_{kj} + e_{ij} \tag{1}$$

For each location, we opted for a three-factor analysis, assuming one for each meaningful source in the area: wet oil and gas activity, dry oil and gas activity, and miscellaneous, which could represent emissions from traffic or natural sources (i.e., trees) depending on the location. Note that separate analyses were conducted for each site, and the species within each group varied depending on the reference pollutants available at each site. Missing values and those below the model detection limit were replaced with half the detection limit for analysis purposes. While uncertainties were provided for the FRAPPE data, an uncertainty of 15% of each measurement was assumed for the CDPHE data.

## 3. Results

### 3.1. Calibration, Best-Fitting ANN Parameters

In developing sensor calibrations for each pollutant using ANNs, we found that varying the inputs to the ANN, such as the sensors, interactions between sensors, and electrical signals (e.g., voltage, resistance), was just as impactful as parameter tuning. Interestingly, raw voltage produced the best fits for all the pollutants studied except for Greeley's biogenic factor pollutant group, which had the lowest reference concentrations. The ratio of the calculated resistance to the resistance of the sensor in clean air, referred to as $R/R_0$, worked the best for this group. The sensor manufacturer recommends using $R/R_0$ in all cases [34,35], yet it was only helpful here when the concentrations were the furthest below the prescribed detection limit. This suggests that the term may not be needed in general. This has been demonstrated in other studies [2,36], but may be useful in cases such as this where the desired concentration is much lower than the sensors are designed to quantify.

Note that several different species use the exact same ANN input data, similarly to how the same linear multivariable equation has been used for both methane and TNMHCs in the past [1,2]. However, the same pollutants or groups of pollutants at the two different sites typically had different inputs and parameters that produced the best results, likely due to the different sets of compounds comprising each group at the two sites. For instance, one group contained benzene and toluene, the only pollutants available at both sites. However, since the rest of the compounds in the oil and gas groups varied, their ANN input signals and parameter tuning were different. Similarly, the remaining two factors entirely comprised different compounds at each site, so the inputs and tuning were understandably mixed. Even when studied alone, benzene and toluene required different ANNs at each site, which may have been influenced by the different time averaging at each, as the minutely FRAPPE data were inherently more precise than the hourly CDPHE data. The inputs and parameters for each species or groups of species are shown in Table 1.

The sensor manufacturer lists several target gases for each of the VOC sensors used. Only two of our target gases made appearances in their literature: isobutane is listed for the light VOC sensor ($VOC_1$), while toluene is listed for the heavy VOC sensor ($VOC_2$) [34,35]. While we will discuss isobutane in Section 3.4, it is worth noting that both toluene models performed best with the addition of a heavy VOC sensor interaction term, which is in line with the manufacturer's usage recommendations. Likewise, the oil and gas group contained toluene, and thus, an additional VOC interaction term was required as well. The term used here represents the ratio of heavy VOCs to total VOCs.

### 3.2. Calibration, PMF Analysis to Group VOCs

Based on our three-factor analysis, factor fingerprints were determined for each species, i.e., what percentage of their concentrations could be attributed to each source. Then, we placed each compound into one of the three groups based on whether at least roughly 60% of the species concentration could be explained by that factor. The factor fingerprints are shown in the right panels of Figure 2, and the final group selections on the left. The constituents of each group were summed to create three distinct groups, and each group was then used to generate ANN calibration models.

**Table 1.** ANN inputs and parameters per location and pollutant. For the sensor inputs, t is temperature in degrees Celsius, h is absolute humidity, $VOC_1$ and $VOC_2$ are the two VOC sensors, $CO_2$ is the $CO_2$ sensor, and Te is the elapsed time.

| Dataset | Species | Electrical Input | Sensor Inputs | Hidden Layers | Hidden Layer Size | Backpropagation Method | Epochs | Performance Goal | μ |
|---|---|---|---|---|---|---|---|---|---|
| FRAPPE | Oil and Gas | Voltage | t, h, $VOC_1$, $VOC_2$, Te, $VOC_2/(VOC_1 + VOC_2)$ | 4 | 5 | Bayesian Regularization | 1000 | $1 \times 10^{-10}$ | 0.5 |
| | Combustion | Voltage | t, h, $VOC_1$, $VOC_2$, Te, $CO_2$, $VOC_1/(VOC_1 + VOC_2)$ | 1 | 10 | Levenberg-Marquardt | 100 | $1 \times 10^{-10}$ | 0.01 |
| | Natural Gas | Voltage | t, h, $VOC_1$, $VOC_2$, Te, $CO_2$, $VOC_1/VOC_2$, $\ln(t) \times h$ | 1 | 10 | Levenberg-Marquardt | 100 | $1.00 \times 10^{-10}$ | 0.1 |
| | Benzene | Voltage | t, h, $VOC_1$, $VOC_2$, Te, $CO_2$ | 1 | 10 | Levenberg-Marquardt | 100 | $1 \times 10^{-10}$ | 0.005 |
| | Toluene | Voltage | t, h, $VOC_1$, $VOC_2$, Te, $VOC_1/VOC_2$, $VOC_2$ t | 4 | 7 | Bayesian Regularization | 1000 | $1 \times 10^{-10}$ | 0.5 |
| | HCHO | Voltage | t, h, $VOC_1$, $VOC_2$, Te, $CO_2$, $VOC_1/(VOC_1 + VOC_2)$ | 1 | 10 | Levenberg-Marquardt | 100 | $1 \times 10^{-10}$ | 0.005 |
| Greeley | Oil and Gas | Voltage | t, h, $VOC_1$, $VOC_2$, Te, $CO_2$, $VOC_2/(VOC_1 + VOC_2)$ | 1 | 10 | Levenberg-Marquardt | 100 | $1 \times 10^{-10}$ | 0.05 |
| | Natural Gas | Voltage | t, h, $VOC_1$, $VOC_2$, Te, $CO_2$, $VOC_1/VOC_2$ | 1 | 5 | Bayesian Regularization | 1000 | 0.01 | 0.5 |
| | Biogenic | $R/R_0$ | t, h, $VOC_1$, $VOC_2$, Te, $VOC_2/(VOC_1 + VOC_2)$, $VOC_1$ t | 1 | 5 | Bayesian Regularization | 1000 | 0.01 | 0.5 |
| | Benzene | Voltage | t, h, $VOC_1$, $VOC_2$, Te, $CO_2$, $VOC_1/VOC_2$ | 1 | 10 | Levenberg-Marquardt | 100 | $1 \times 10^{-10}$ | 0.005 |
| | Toluene | Voltage | t, h, $VOC_1$, $VOC_2$, Te, $VOC_1/VOC_2$, $VOC_2$ t | 1 | 10 | Levenberg-Marquardt | 100 | $1 \times 10^{-10}$ | 0.01 |

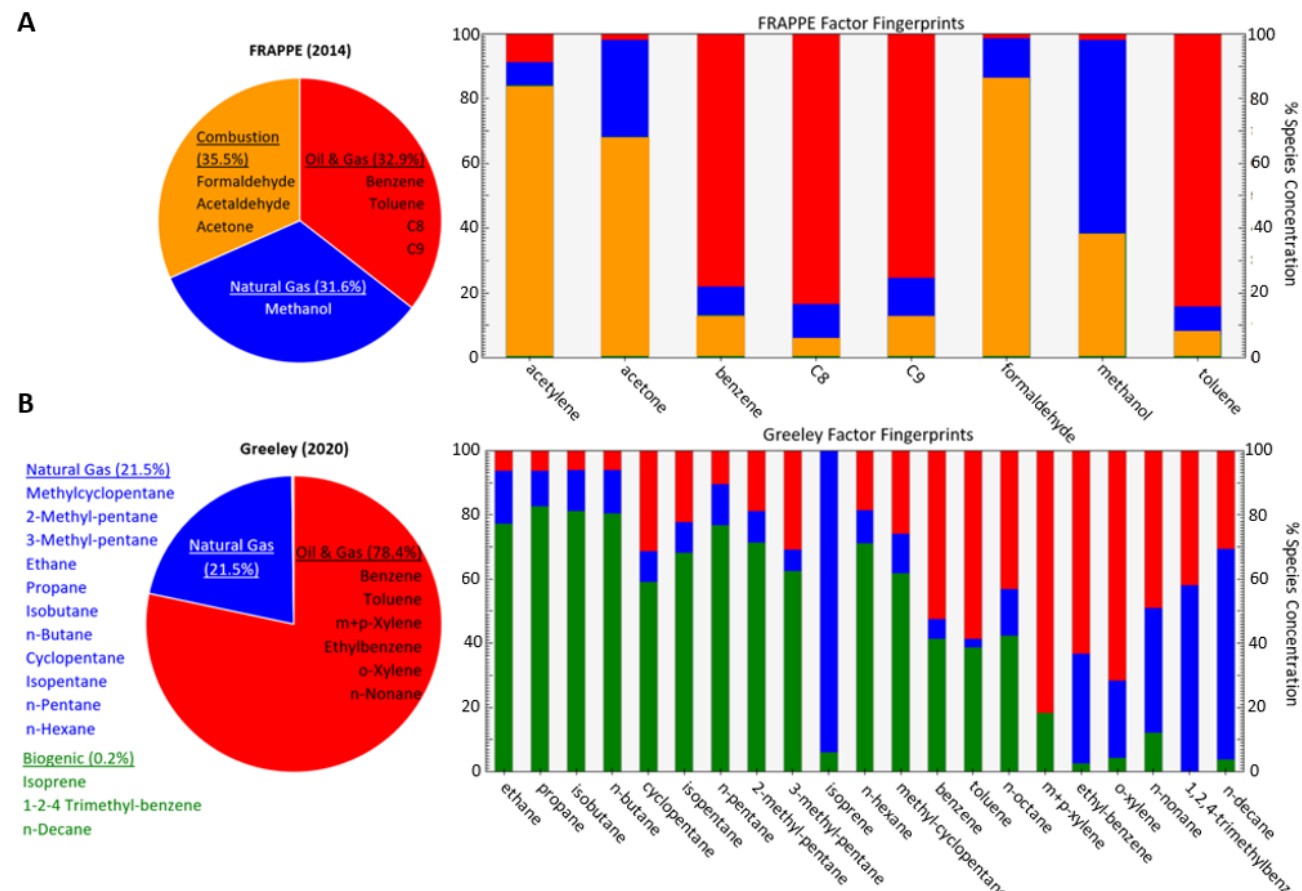

**Figure 2.** Individual pollutant contributions and factor fingerprints for FRAPPE (**A**) and Greeley (**B**).

For the FRAPPE data, one factor ("FRAPPE oil and gas") consisted of the aromatic compounds, benzene, toluene, $C_8$, and $C_9$, and was considered the aromatics or BTEX group. While these compounds may be emitted from traffic [37] based on the location, we

are considering this primarily as the liquid phase of oil and gas emissions [38], although both sources likely contributed. The next group ("combustion") consisted of aldehydes (formaldehyde and acetaldehyde) and acetone, which are all generated through fossil fuel combustion processes [39]. While traffic could have contributed to combustion concentrations, flaring is the more likely source in this area, and these represent gas-phase oil and gas emissions. Lastly, the final group ("FRAPPE natural gas") contained methanol alone, which is likewise emitted from natural gas [40]. It is important to note that the eight hydrocarbons available for analysis at this site were far from exhaustive, and these were selected for analysis by NASA due to their roles as ozone precursors [41]. Although other compounds were undoubtedly emitted from other sources, we are only able to characterize these select few, most of which have links to oil and gas activity. Thus, all three of our sources are connected to oil and gas, while biogenic and other emissions expected in a rural area have been overlooked.

At CDPHE's reference site in Greeley, a much larger panel of 21 hydrocarbons was available. While this is still not an exhaustive selection of every possible hydrocarbon emitted in this area, the reference data afforded us a more complete understanding of sources in the area besides those associated with natural gas. Similar to FRAPPE, one group ("Greeley oil and gas") comprised the full set of aromatics or BTEX compounds, including benzene, toluene, ethylbenzene, and multiple xylenes, in addition to one other compound, n-nonane, primarily representing liquid-phase natural gas emissions [42]. Another group ("Greeley natural gas") includes a large variety of alkanes, many of which are the components of natural gas and also represent oil and gas emissions [43]. The final factor ("biogenic") represents biogenic compounds such as isoprene, which is emitted from plants [44]. Note that n-octane appeared to be evenly split between the oil and gas and natural gas groups; since neither held a majority, it was not included in the remainder of our analyses to avoid overlap among the separate concentrations we fit neural networks to. For clarity, each of these groups will be referred to as its likely source from this point forward, e.g., the aromatic groups will be referred to as "oil and gas". Bear in mind that the two groups present at both sites ("oil and gas", "natural gas") represent entirely different sets of compounds at each site, sharing their name only due to PMF-aided source attribution rather than chemical structure.

### 3.3. Calibration for Individual VOCs, Benzene and Toluene

Although the species available for analysis at the two locations varied widely, two compounds were available at each: benzene and toluene. The comparison of the calibration fit for each, at each site, is shown in Figure 3; the quality of the fit may be indicative of the influence of time resolution on calibration. Both compounds displayed better fits at Greeley, where hourly data were available, than at the FRAPPE site, for which minute-average data were used. A previous study showed that higher time averaging and, therefore, sparser data points can cause calibration models to overfit [36]. However, in this case, the large spikes of both compounds may have contributed more to the poor FRAPPE fits rather than the time averaging. When the minutely data from FRAPPE were hourly averaged, the fit for benzene improved only slightly (calibration $R^2 = 0.30$, validation $R^2 = 0.22$). Likewise, when the FRAPPE toluene data were hourly averaged, the calibration and validation folds showed less variability, but fit worse overall (calibration $R^2 = 0.38$, validation $R^2 = 0.36$). This suggests that the time averaging used is not the reason for the different fits for the same pollutants at the two different sites.

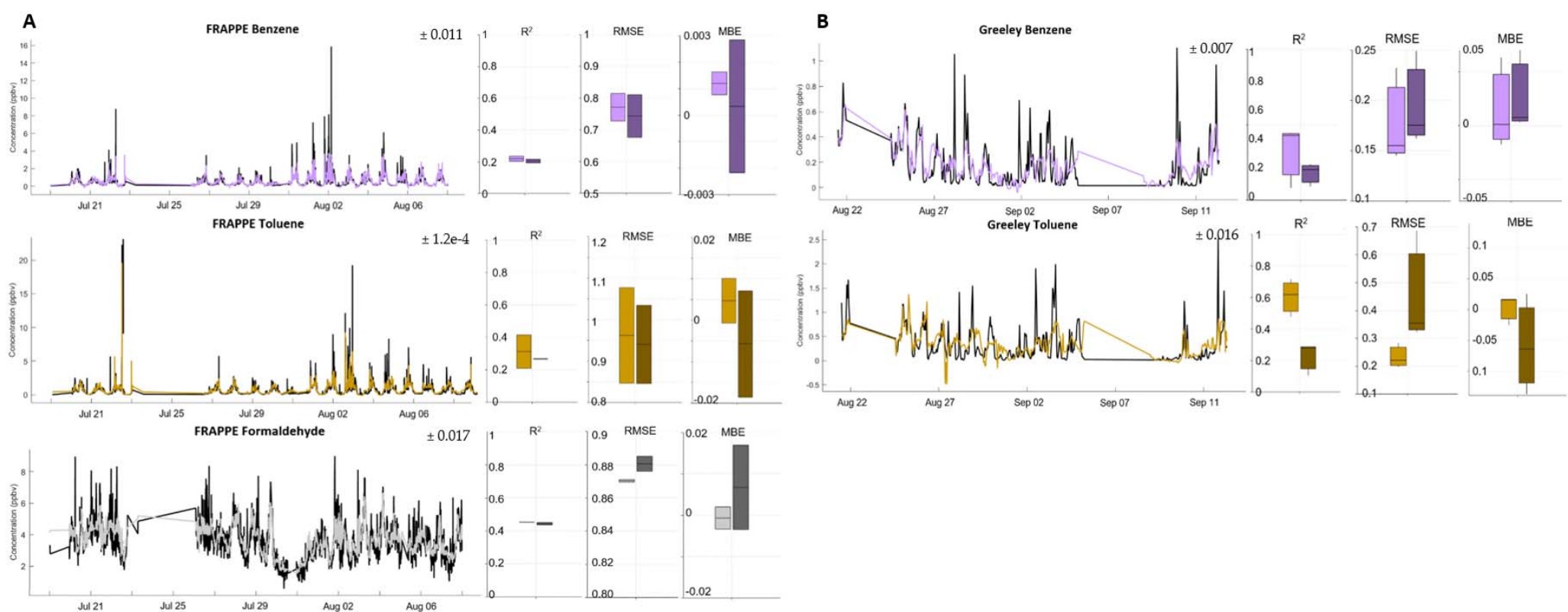

**Figure 3.** FRAPPE (**A**) and Greeley (**B**) speciated colocation timeseries (colored; reference data in black) along with boxplots for folds of calibration (light) and validation (dark): R$^2$, random error (root mean square error, or RMSE), and bias (mean bias error or MBE). Estimate uncertainties are overlayed on the timeseries.

The key similarity between the fits for these two individual species, as well as the rest of the pollutants studied, was the ability of the sensors to capture baseline trends in the reference concentration data while struggling to model intermittent peaks. The FRAPPE reference data for these two compounds display more frequent spikes of relatively high concentrations as compared to the rest of the data collected. Since all our models struggled with peak finding, the lower $R^2$ values and random error are likely being driven primarily by the consistent underpredicting of large peaks.

Without using additional reference signals as in Collier-Oxandale et al. [21], the median calibration and validation fits for benzene were significantly reduced. While the FRAPPE benzene and toluene ANN fits need further refinement to be reliable enough for wider use, the promising fits seen at Greeley show that this method may be a strong enough start for quantifying speciated VOCs of interest.

To ensure the significance of our results, we tested the null hypothesis that the median Pearson's R across all folds of calibration data for each compound was not equal to zero at the 0.05 significance level. For the FRAPPE data, based on minutely averaged samples ($20,327 \leq n \leq 23,425$ for each dataset), all compounds and groups of compounds were found to be statistically significant. For the Greeley data, which were hourly averaged ($n = 371$ for each dataset), all compounds and groups of compounds were likewise found to have R values statistically different from zero.

### 3.4. Calibration of Grouped VOCs Using PMF

As expected, since the summed species in the groups had higher overall concentrations than their constituents alone, fits were generally better. Since the groups comprised different compounds at each site, this is not a clear-cut comparison but rather a discussion of two separate sets of quasi-speciated compounds. The correlations among groups at each site were first examined (see Supplementary Figure S1) to ensure that consistencies among fits were not merely caused by similarities in the input data.

As seen in Figure 4, the species with the fewest and smallest spikes had the best fits, as all our models struggled to capture peaks in the reference concentration data. FRAPPE's combustion and natural gas groups had slightly higher concentration ranges than other compounds and saw very few spikes above baseline, and as such, the models were able to capture more than half of the variability in the pollutants. FRAPPE's oil and gas data, which included benzene and toluene, likely struggled due to the same issues that plagued these two pollutants on their own.

Some groups at Greeley fit better than others, but overfitting was an issue across the board, perhaps due to the hourly time averaging resulting in fewer data points in the ANN model development (~400 points per pollutant) as compared to FRAPPE (~1000+). The Greeley oil and gas group, which also contained benzene and toluene, displayed fits not unlike those seen for its individual constituents. Since the natural gas and biogenic groups were entirely different from their FRAPPE counterparts, our results varied and do not provide a clear comparison. The Greeley natural gas group represents a host of alkanes and experienced significantly more bias than any other compound or group of compounds, suggesting that the sensors themselves may have struggled to pick up on compounds in this group. Although isobutane, one of the members of this group, is listed by the manufacturers as a target gas for the light VOC sensor, reference concentrations for the individual compound had a median of 1.5 ppb, and the compound was one of 11 included in the group; concentrations were likely not significant enough to be singled out by the sensor.

The biogenic group experienced the worst calibration and validation fits out of all at Greeley. Based on the PMF analysis, this group comprised a mere 0.2% of all the total VOCs quantified; it was not a major contributor, and the small concentrations there were difficult to fit. The highest spikes for the biogenic group's three compounds (isoprene, 1-2-4-trimethyl-benzene, and n-decane) combined totaled less than 1 ppb, so it is unsurprising that this group had the least successful fits.

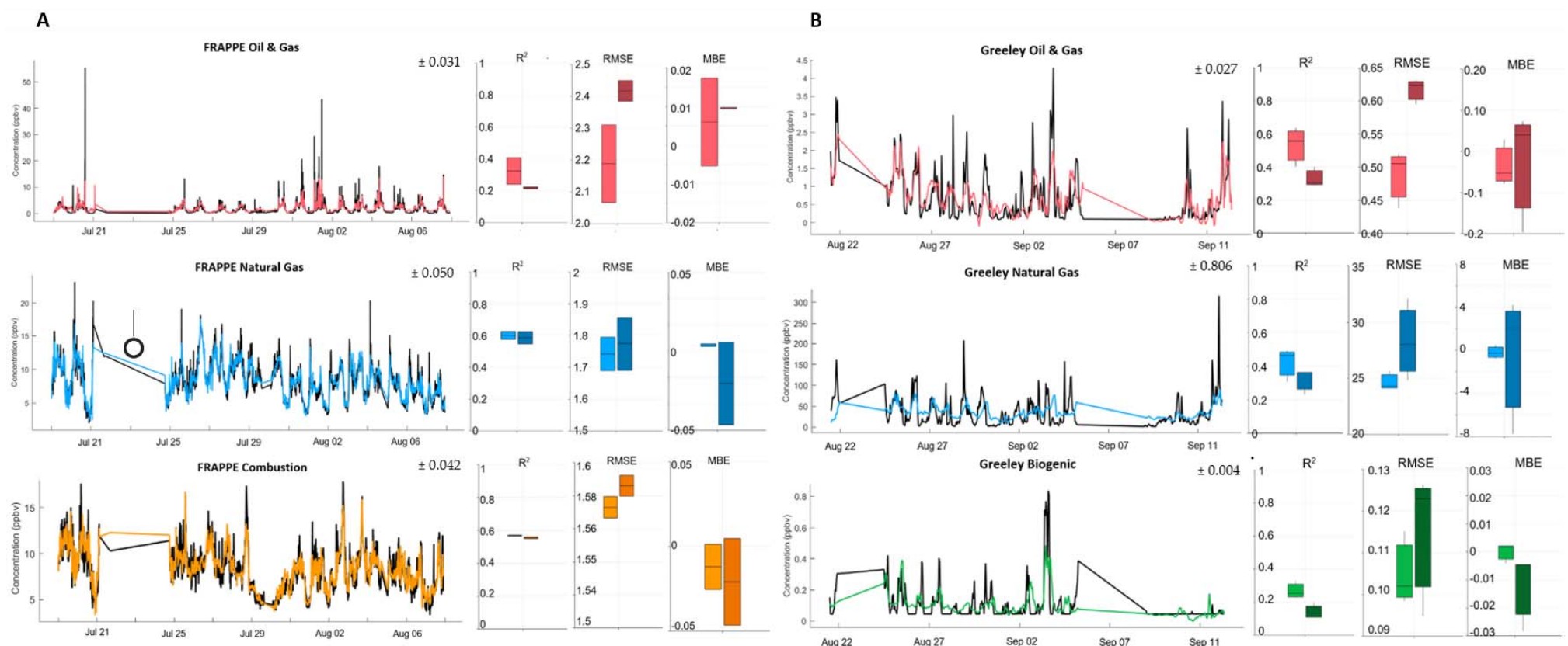

**Figure 4.** FRAPPE (**A**) and Greeley (**B**) source-apportioned colocation timeseries (colored; reference data in black) along with boxplots for folds of calibration (light) and validation (dark): $R^2$, random error (root mean square error, or RMSE), and bias (mean bias error, or MBE). Estimate uncertainties are overlayed on the timeseries.

### 3.5. Calibration of Formaldehyde

As a case study, we implemented ANN fits for one additional species only available at the FRAPPE site, formaldehyde, which was included in the natural gas group at the site (aldehydes). This species was of particular interest due to its role in acid rain formation [45] and toxicity to humans and animals [46]. Due to the lack of large spikes in the data, this compound's calibration and validation fits hovered just below 0.5 for $R^2$, with our model explaining approximately half the variability in the compound. Bias and random error were also relatively minor. Since formaldehyde was only available during the FRAPPE campaign, we cannot draw comparisons to any other datasets. However, our ability to model the compound at this one site is promising for future applications. These results are shown in Figure 3.

### 3.6. Application to Field Data

As a case study, we applied the calibration ANNs generated for the oil and gas and natural gas groups at the Greeley reference site to field data collected in Wiggins, CO, USA, as marked in Figure 1. This step is crucial as the calibration data studied are sometimes prone to overfitting, and likewise, small differences in environmental conditions (i.e., temperature, humidity) between co-location and deployment phases may contribute to less-than-ideal fits in the field. The pod was placed approximately 3 m west of oil and gas equipment; note that the prevailing wind direction in the area was northerly at the time. We elected to apply the fits for these two groups only for several reasons. Firstly, since concentrations of individual species were estimated to be very small in this area, using the groups of multiple species summed would allow for slightly higher concentrations to be fitted. Secondly, with the oil and gas factor representing BTEX compounds, this might represent liquids present close to oil and gas equipment [47]. Likewise, the natural gas group might represent gases associated with the pumping and metering equipment mere meters from the pod [48–50]. The results of each are shown in Figure 5.

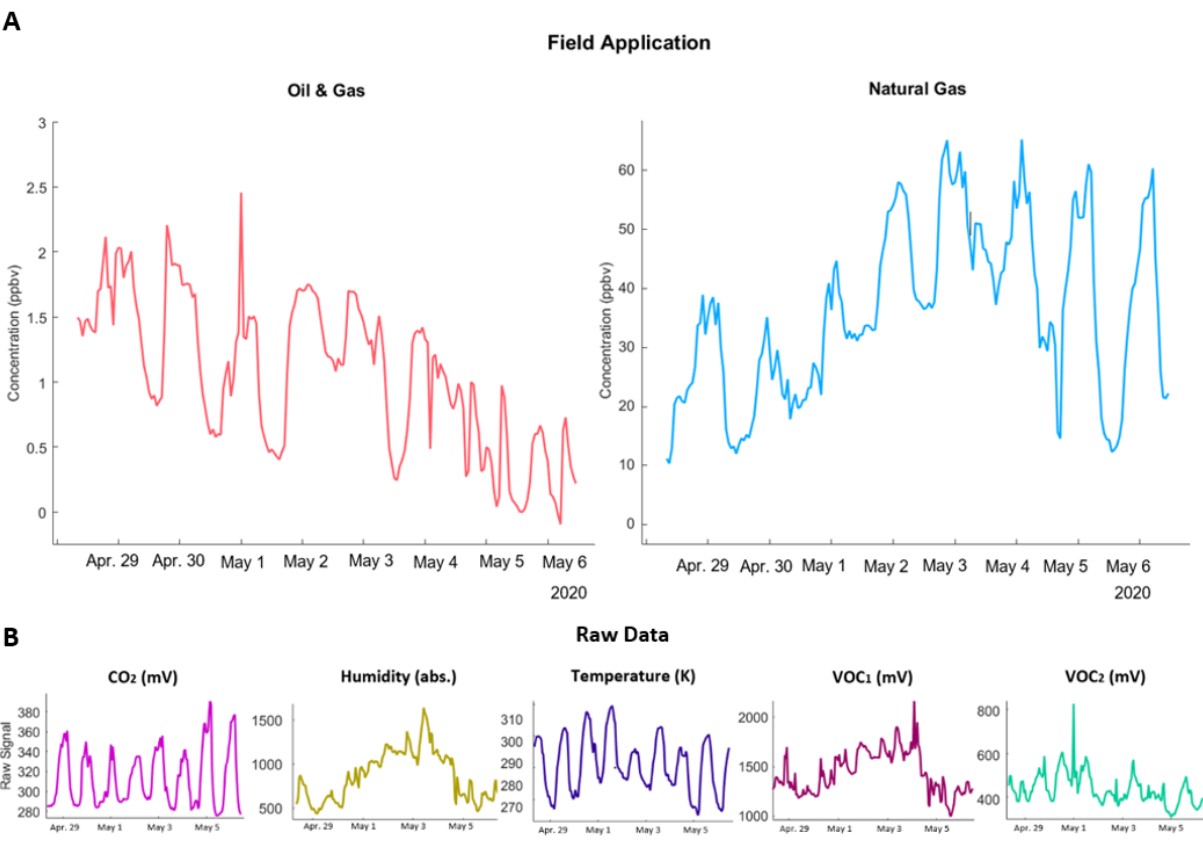

**Figure 5.** Field applications of the ANNs (**A**); raw pod data used to generate these estimates (**B**).

While most trace gases studied demonstrate a diurnal effect where concentrations rise and fall as the boundary layer changes height throughout the day [51], these quasi-speciated compounds rise during the day, but return to different baselines each night. This may have been influenced by the colocation data, since the reference data for benzene exhibited some baseline shifting throughout the week of colocation (see Figure 4).

Upon examining the raw data collected by the pods, substantial baseline shifting is observed across all sensors. Large swings in temperature and humidity are typical of the spring season in Colorado, but given the remaining sensors' sensitivity to temperature and humidity, may have had a larger effect on results than anticipated. The light VOC sensor ($VOC_1$) in particular appears to mirror the trends in humidity, suggesting that the increasing light VOC baseline was due to environmental conditions rather than the presence of pollutants. While the $CO_2$ and heavy VOC ($VOC_2$) sensors exhibit more typical patterns, they likewise fail to return to the same baseline each night.

These trends in the raw data suggest that the drift experienced when applied to the field data was primarily driven by unstable environmental conditions, which in turn affected each of the sensors. In future analyses, we will aim to deploy in environments with more stability to better characterize the utility of the ANN method independent of unforeseen driving factors. Nevertheless, this analysis demonstrates that speciated VOC ANNs may be suitable for longer-term environmental monitoring, although baseline shifting techniques may still be necessary for reliability in the future. Concentration ranges are reasonable given the location, and typical diurnal patterns are still observed regardless of baseline drift.

## 4. Discussion

### 4.1. Comparison of Two Locations

Several key differences between the two datasets from the two sites may have driven the inconsistencies between the fits for each pollutant or group. The observed discrepancies were driven primarily by the species available at each site rather than differences in the oilfield, production, or other relevant nearby sources. Likewise, the differences in fits between the two may have been driven by the metal oxide sensors' ability to recognize the different pollutants as the concentrations at each site did not vary widely (see Figure 6). Different concentrations and time averaging may have also contributed to fits across the two sites being less than consistent, as hourly data are prone to overfitting, which explains the worse validation fits at the Greeley site.

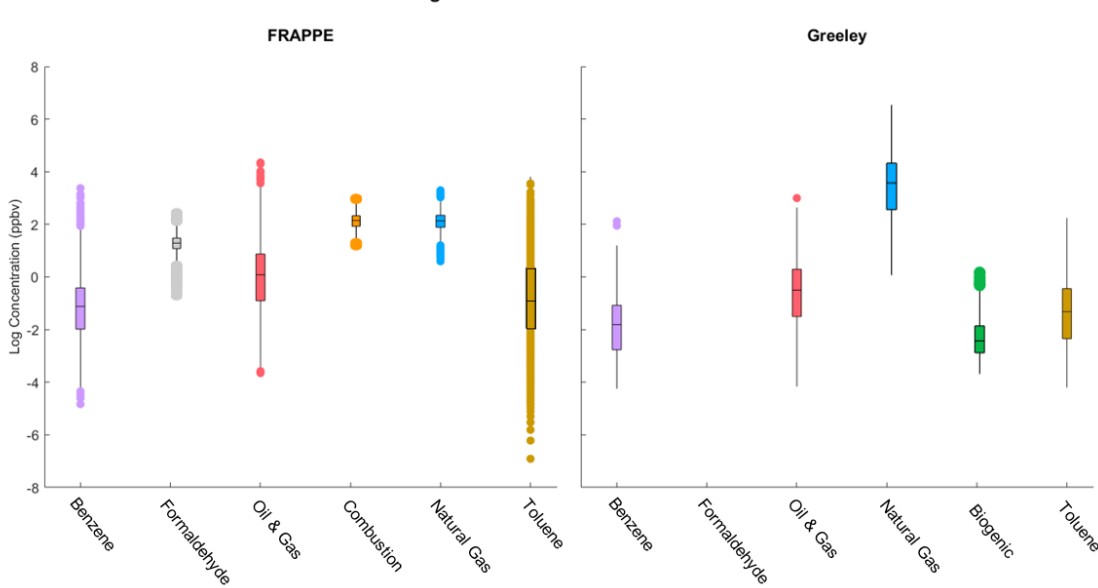

**Figure 6.** Reference concentrations of individual and grouped pollutants by site.

Some consistencies among the two sites and different sets of pollutants highlight the strengths and weaknesses of our methods. While baseline trends were generally captured, larger spikes were harder for the ANNs to model, resulting in worse fits for compounds experiencing spikes and better fits for those without.

### 4.2. Broader Implications

VOC source attribution has often been muddled due to the large range of compounds encompassed by the TNMHC category. In a previous low-cost sensor work concerning oil and gas development in Los Angeles, CA, USA, it was not abundantly clear whether hydrocarbon emissions were emanating from oil and gas facilities or roadway traffic—or to what extent each played a role [2]. In knowing the exact makeup of VOC emissions—or groups whose source can be deduced—VOC readings from low-cost sensors will be much more impactful in source attribution.

The implications of determining speciated and quasi-speciated hydrocarbons from more widely distributed sensors cannot be understated, as VOCs are a major precursor to ozone. Recent studies have attempted to model the sensitivity among VOCs, nitrogen oxides ($NO_x$), and ozone to better characterize the non-linear reactions generating ozone [52,53]. The ability to quantify a variety of speciated hydrocarbons at different points in an area of interest, especially near roadways, could illuminate why reducing concentrations of precursors does not necessarily reduce ozone formation [13].

Furthermore, knowing the exact makeup of hydrocarbon emissions will greatly improve our ability to estimate the associated health and environmental risks. The BTEX compounds, which have been linked to oil and gas emissions, are known to be toxic and in some cases carcinogenic [5,6]. Many other VOCs have demonstrated links to adverse health effects, such as eye irritation, headaches, respiratory inflammation, and the exacerbation of pre-existing conditions such as asthma [54,55]. Children, pregnant people, and the elderly face particularly high risks of having adverse health reactions to elevated VOC concentrations from oil and gas [56–58]. In Greeley, our data were collected on the grounds of an elementary school near oil and gas activities; even the small concentrations detected here have the potential for health implications for sensitive groups [59]. Thus, understanding the composition of hydrocarbon emissions in this area is of great importance for assessing health risks. Likewise, certain hydrocarbons are stronger climate forcers than others [60]; the better our assessment of the associated risks, the more impactful our mitigation strategies can be.

### 4.3. Limitations and Lessons Learned

The most pressing shortcoming of this work is the detection limits associated with the metal oxide sensors used. Figaro states their lower detection limit as 1 ppm [34,35], which is orders of magnitude larger than any of the reference concentrations seen during our data collection. While our method is still able to explain a portion of the variability of each compound, higher pollutant concentrations would make for more reliable results and presumably better ANN fits. Our sensor deployment nearby oil and gas was not the most suitable based on the concentration ranges seen, but this approach might be better suited for fence line monitoring where higher baseline concentrations might be observed [2]. Individual VOC emissions rarely top this threshold, but other applications with higher concentrations than those observed here such as wildfire plumes [61] might be more acceptable. However, the issues surrounding spike finding would need to be addressed to use this method successfully.

Another major limitation is the difficulty our models had in representing concentration spikes. While our baseline fits appear reliable for each of the compounds studied, large spikes are typically of greater interest to community members based on the associated health effects. While we continue to develop methods to quantify spikes specifically, in the meantime, we will consider any measurements above baseline concentration as spikes for community monitoring purposes. We may not be able to adequately measure exact

spike concentrations at the present, but being able to warn stakeholders of the presence of a spike may be sufficient until a better method is achieved.

It is worth noting that higher concentrations may only be required during the colocation phase of the deployment to attain better ANN fits. Based on prior pod research, assuming a decent colocation is carried out, calibration fits can be applied to field deployments in varying locations with vastly different temperature, humidity, and concentration spaces [2]. Thus, the application may not need to have abnormally high VOC concentrations so long as those experienced during the colocation were sufficient to achieve reasonable ANN fits. Perfect results will not be achieved with sensors developed neither for this purpose nor concentration range, but these results represent an important first step in making speciated hydrocarbon data available in ambient outdoor monitoring. Based on the utility of VOC sensor interaction terms in our ANN models, we believe that the use of additional VOC sensors may further our ability to quantify specific hydrocarbons or hydrocarbons from specific sources in the future. In addition to the generalized "heavy VOC" and "light VOC" sensors, a mid-range VOC sensor or one specific for aromatics, for instance, might bolster our abilities even further.

## 5. Conclusions

While this method is imperfect, it has great potential for researchers to quantify specific hydrocarbons on highly localized spatial and temporal resolutions. Source mixtures with low variability or a dominant source appear best suited for this method; additional measures will likely be needed to quantify less-than-ideal mixtures. Baseline concentrations developed using our ANN calibration method appear reliable in some cases, while spike finding will prove to be more challenging. PMF may be a valuable tool in connecting individual compounds to emission sources when their concentrations are too small to be studied individually. Recent studies have quantified TNMHCs with $R^2$ values hovering around the 0.5 threshold [1,2] for outdoor experiments using a similar field normalization technique. Our ability to now quantify specific hydrocarbons rather than all these lumped together to a similar degree of accuracy represents an improvement in the field and may be further bolstered by low-cost separation techniques upstream of sensors in the future. Continuing to look forward, this type of tool will better inform us of the human exposure and environmental hazards posed by hydrocarbon emissions.

**Supplementary Materials:** The following are available online at https://www.mdpi.com/article/10.3390/atmos12111383/s1, Figure S1: Colored correlation matrices for FRAPPE & Greeley PMF groups.

**Author Contributions:** Conceptualization, M.H.; Formal analysis, K.O.; Visualization, K.O.; Writing—original draft, K.O.; Writing—review & editing, M.H. All authors have read and agreed to the published version of the manuscript.

**Funding:** Publication of this article was funded by the University of Colorado Boulder Libraries Open Access Fund.

**Data Availability Statement:** Reference data is available online at https://www-air.larc.nasa.gov/cgi-bin/ArcView/discover-aq.co-2014?GROUND-PLATTEVILLE=1 (see data from Armin Whisthaler, 21 October 2021) and https://www.colorado.gov/airquality/tech_doc_repository.aspx (see "Extraction Vetting (Bella Romero)", 21 October 2021) for FRAPPE and Greeley respectively. Y-Pod data is available upon request; please contact the first author.

**Acknowledgments:** Thank you to CDPHE and all those involved in the NASA FRAPPE campaign, especially Ashely Collier-Oxandale and Hannah Halliday. Big thanks to Evan Coffey and Daniel Bonn for their help with the pod colocations.

**Conflicts of Interest:** The authors declare no conflict of interest.

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
