# Peer review of "Applications and Limitations of Quantifying Speciated and Source-Apportioned VOCs with Metal Oxide Sensors"

_atmosphere, doi:10.3390/atmos12111383_

Round 1
Reviewer 1 Report
This research article by K. Okorn and M. Hannigan deals with the study of “Applications and limitations of quantifying speciated and source apportioned VOCs with metal oxide sensors”. Manuscript is well-organized and the authors explained have several facts, along with drawbacks, of the work properly. The results of this study represent an important first step in making specialized hydrocarbon data available in ambient outdoor monitoring, which will be valuable to the sensor community. Therefore, this paper is acceptable in its present form and no further revisions are required. Just one minor suggestion is that, the authors have explained the shortcomings of the detection limit with metal oxides. So, authors should comment in the conclusion section (that can be done during proof-reading) that how to design and prepare such materials that can detect low level target gases with high accuracy.Author Response
Manuscript Atmosphere- 1388417
Response to Reviewer 1
Ms. Zheng,
Thank you for the opportunity to submit a revised draft of the manuscript titled, “Applications and limitations of quantifying speciated and source apportioned VOCs with metal oxide sensors”. We truly appreciate the time and effort that you and the reviewers devoted to reading and providing feedback on our manuscript. We are grateful for your suggestions and the improvements that they have brought to this paper. Please find a point-by-point response of how we incorporated the reviewers’ comments and concerns into this draft.
Author’s Comments to Reviewer #1:
Reviewer #1 Synopsis: This study is one of the first steps in speciating hydrocarbon data for ambient outdoor monitoring, which is an important improvement for the sensor world.
Author Response: Thank you for your comments. We are glad that the main points of our paper were well-received.
- Reviewer Comment: Authors should comment in the conclusion section (that can be done during proof-reading) that how to design and prepare such materials that can detect low level target gases with high accuracy
Author Response: Thank you for mentioning this. We have proofread the article in full for minor spelling and grammatical errors, and have added mention of another method (separation techniques) with potential for future use in the conclusion section.

Reviewer 2 Report
The work addresses an actual issue of possibilities and limitations of quantifying speciated and grouped VOCs with metal oxide sensors. The interest in applying sensors for ambient air monitoring is on rise, especially in the vicinity of industrial sources. The work is interesting, however the presentation of the results raises some controversy.
The R2 for ANN models representing benzene and toluene are so low, that their statistical significance is questionable. Their significance should be explicitly tested and revealed. The obtained results provide an argument against applying metal oxide sensors for quantifying speciates VOCs, at least in real environmental conditions. However, the discussion and conclusions in the paper go in the opposite direction. In principle poor results of individual species quantization are not a surprise. Semiconductor gas sensors are semi-selective and they are dedicated to measuring wide range of compounds in total. Authors confirm this, at least partially, by showing good results for FRAPPE ‘natural gas’ and ‘combustion’ categories. In my opinion the overall message of the paper is different than the one articulated and this should be changed.
The proposal to quantify groups of VOCs by source apportionment is interesting. However, due to different scope of analytical measurements, resulting in different sets of compounds which belong to the same groups e.g. ‘oil and gas’ at different locations the presented results are misleading. No general conclusions can be drawn regarding this kind of sensors application. Please consider other possibility of grouping which would allow for such conclusions.
Fig. 2 is not readable. Please improve.
Fig. 5 – it is not clear what this figure is meant to demonstrate.
Author Response
Manuscript Atmosphere- 1388417
Response to Reviewer 2
Ms. Zheng,
Thank you for the opportunity to submit a revised draft of the manuscript titled, “Applications and limitations of quantifying speciated and source apportioned VOCs with metal oxide sensors”. We truly appreciate the time and effort that you and the reviewers devoted to reading and providing feedback on our manuscript. We are grateful for your suggestions and the improvements that they have brought to this paper. Please find a point-by-point response of how we incorporated the reviewers’ comments and concerns into this draft.
Author’s Comments to Reviewer #1:
- Reviewer Comment: The statistical significance of the R2 values for benzene and toluene should be tested and stated.
Author Response: Thank you for pointing this out. We have added the uncertainty estimates for all compounds directly to figures 3 and 4 for clarity.
- Reviewer Comment: The results show that metal oxide sensors should not be used to quantify speciated VOCs, as further evidenced by the improved categorized fits (e.g. FRAPPE ‘natural gas’ and ‘combustion’). The overall message of the paper should be updated to reflect this.
Author Response: Thank you for bringing this to our attention. We did not mean to overstate our results as a widespread method for VOC speciation, but rather as a first step with room for growth in using low-cost sensors in this aim. In response, we have updated mentions of this in the introduction and conclusion sections to stress the applications where this approach appears reasonable (source mixtures with low variability or one dominant source present) to make this clearer.
- Reviewer Comment: Labeling different sets of compounds as being part of the same groups at different locations is misleading and conclusions cannot be drawn from this. Consider other grouping possibilities to allow for these conclusions.
Author Response: Thank you for your comment. We have added an additional explanation of why the groups are named as such; despite having different mixtures, the PMF analysis indicated that they were emanating from the same sources. We have also updated several instances of “oil and gas” or “oil and gas at Greeley” to “Greeley oil and gas” to make distinctions between similarly named groups clearer, and likewise for FRAPPE and “natural gas”.
- Reviewer Comment: Figure 2 is not readable.
Author Response: Thank you for bringing this to our attention. We have improved the font sizes on each component of the figure.
- Reviewer Comment: The meaning of Figure 5 is unclear.
Author Response: Thank you for your comment. We have updated the text to better describe why a field application has been shown, as even models with acceptable calibration data sometimes struggle to perform once actually deployed due to overfitting or small changes in environmental conditions.

Round 2
Reviewer 1 Report
Revised version is acceptable.
Reviewer 2 Report
Thank you for responding to my comments.
ad. 1 Please provide the p-values for each R2. The uncertainties you quoted in fig 3 and fig 4 are redundant, as they were not explained in figure captions and they were not referred in the text of the manuscript. The objective indication of the statistical significance of the developed models is still missing.
ad. 5 In Fig. 5, please present the measurement data and the ANN modeling results jointly.
